# Certified Defense to Image Transformations via Randomized Smoothing

**Marc Fischer, Maximilian Baader, Martin Vechev**
Department of Computer Science
ETH Zurich
{marc.fischer, mbaader, martin.vechev}@inf.ethz.ch

## Abstract

We extend randomized smoothing to cover parameterized transformations (e.g., rotations, translations) and certify robustness in the parameter space (e.g., rotation angle). This is particularly challenging as interpolation and rounding effects mean that image transformations do not compose, in turn preventing direct certification of the perturbed image (unlike certification with $\ell^p$ norms). We address this challenge by introducing three different defenses, each with a different guarantee (heuristic, distributional and individual) stemming from the method used to bound the interpolation error. Importantly, in the individual case, we show how to efficiently compute the inverse of an image transformation, enabling us to provide individual guarantees in the online setting. We provide an implementation of all methods at https://github.com/eth-sri/transformation-smoothing.

## 1 Introduction

Deep neural networks are vulnerable to adversarial examples [1] – small changes that preserve semantics (e.g., $\ell^p$-noise or geometric transformations such as rotations) [2], but can affect the output of a network in undesirable ways. As a result, there has been substantial recent interest in methods which aim to ensure the network is certifiably robust to adversarial examples [3–13].

**Certification guarantees** There are two principal robustness guarantees a certified defense can provide at inference time: (i) the (standard) distributional guarantee, where a robustness score is computed offline on the test set to be interpreted in expectation for images drawn from the data distribution, and (ii) an individual guarantee, where a certificate is computed online for the (possibly perturbed) input. The choice of guarantee depends on the application and regulatory constraints.

**Guarantees with $\ell^p$ norms** When considering $\ell^p$ norms, existing certification methods can be directly used to obtain either of the above two guarantees: for an image $\boldsymbol{x}$ and adversarial noise $\delta$, $\|\delta\|_p < r$, proving that a classifier $f$ is $r$-robust around $\boldsymbol{x}' := \boldsymbol{x} + \delta$ is enough to guarantee $f(\boldsymbol{x}) = f(\boldsymbol{x}')$. That is, it suffices to prove robustness of a perturbed input in order to certify that the perturbation did not change the classification, as the $r$-ball around $\boldsymbol{x}'$ includes $\boldsymbol{x}$.

**Key challenge: guarantees for geometric perturbations** Perhaps not intuitively, however, for more complex perturbations such as geometric transformations, proving robustness around an image $\boldsymbol{x}'$ via existing methods (e.g., [9–12]) does not imply that $f(\boldsymbol{x}) = f(\boldsymbol{x}')$ for the original image $\boldsymbol{x}$. To illustrate this issue, consider the rotation $R_\gamma$, by angle $\gamma$ of an image $\boldsymbol{x}$, followed by an interpolation $I$. Certifying that the classification of the rotated image $\boldsymbol{x}' := I \circ R_\gamma(\boldsymbol{x})$ for $\|\gamma\| < r$ is robust under further rotations $I \circ R_\beta$ for $\|\beta\| < r$ is not sufficient to imply that $\boldsymbol{x}$ and $\boldsymbol{x}'$ classify the same, as rotating $\boldsymbol{x}'$ back by $\beta = -\gamma$ does not return the original image $\boldsymbol{x}$ due to interpolation. A central challenge then is to develop techniques that are able to handle more involved perturbations.

**This work: certification beyond $\ell^p$ norms**  In this work we address this challenge and introduce the first certification methods for geometric transformations based on smoothing which provide the above two guarantees. Concretely, we extend randomized smoothing [7] to handle parameterized transformations (SPT) by adding (Gaussian) noise to the parameters of a transformation, enabling us to handle large models and datasets (e.g., ImageNet). We present three methods, BASESPT, DISTSPT and INDIVSPT yielding different guarantees.

**BASESPT**  The guarantees provided by SPT hold for composable parameterized perturbations $\psi_\gamma$, that is $\psi_{\beta+\gamma} = \psi_\gamma \circ \psi_\beta$. SPT can be applied directly to obtain both a distributional guarantee and an individual guarantee. However, if used with non-composable transformations (e.g., rotations with interpolation), BASESPT will yield a heuristic guarantee.

**DISTSPT**  By estimating a probabilistic upper bound for $\epsilon = \|\psi_\beta \circ \psi_\gamma(\boldsymbol{x}) - \psi_{\beta+\gamma}(\boldsymbol{x})\|_2$ offline on the training dataset, SPT can be combined with randomized smoothing (yielding DISTSPT) to provide the standard distributional guarantee for non-composable transformations.

**INDIVSPT**  To obtain an individual guarantee for non-composable transformations, we calculate at inference time for each input $\boldsymbol{x}'$ an individual upper bound of the expression $\epsilon$ *without access* to $\boldsymbol{x}$, which is combined with SPT and smoothing to yield INDIVSPT. A key step here is computing the inverse $\psi_\gamma^{-1}(\boldsymbol{x}')$ of a transformed image $\boldsymbol{x}'$, for which we introduce an efficient technique.

We remark that all three methods are suitable for online use as a defense and the choice of method depends on the particular trade-offs. To summarize, our core contributions are:

- A generalization of randomized smoothing to parameterized transformations.
- A certification method which provides a distributional guarantee by calculating an upper bound on the interpolation error on the training dataset.
- A certification method which provides an individual guarantee based on an algorithm that efficiently calculates the inverse of an image.
- A thorough evaluation of all methods on common image datasets, achieving provable distributional robust accuracy of $73\%$ for rotations with up to $\pm 30°$ on Restricted ImageNet.

## 2  Related Work

We now survey the most closely related work in neural network certification and defenses.

**$\ell^p$ norm based certification and defenses**  The discovery of adversarial examples [1, 14] triggered interest in training and certifying robust neural networks. An attempt to improve model robustness are empirical defenses [15, 16], strategies which harden a model against an adversary. While this may improve robustness to current adversaries, typically robustness cannot be formally verified with current certification methods. This is because complete methods [17–19] do not scale and incomplete methods relying on over approximation lose too much precision [3, 20, 21, 6, 10, 22], even for networks trained to be amenable to certification. Recently, randomized smoothing was introduced, which could for the first time, certify a (smoothed) classifier against norm bound $\ell^2$ noise on ImageNet [23, 24, 7, 8, 25], by relaxing exact certificates to high confidence probabilistic ones. Smoothing scales to large models, however, it is currently limited to norm-based perturbations.

**Semantic perturbations**  Transformations such as translations and rotation can produce adversarial examples [2, 26]. An enumerative approach certifying against semantic perturbations was presented in [9]. There, the search space is reduced by only considering next neighbor interpolation. Unfortunately, for more elaborate interpolations, like bilinear interpolation, this approach becomes infeasible. The first certification against rotations with bilinear interpolations was carried out in [10], which was later significantly improved on by [11]. Both methods generate linear relaxations and propagate them through the network. However, these methods do not yet scale to large networks (i.e., ResNet-50) or complex data sets (i.e., ImageNet). The approaches of [12] is similar to the one in [10] for rotation. The method in [13] is a combination of enumeration and smoothing. The methods for translation in [12] and [13] are in fact equivalent to Pei et al. [9]. Further, [13] can only invoke their method for unperturbed images, making it inapplicable as a defense.

# 3 Generalization of Smoothing

A smoothed classifier $g \colon \mathbb{R}^m \mapsto \mathcal{Y}$ can be constructed out of an ordinary classifier $f \colon \mathbb{R}^m \mapsto \mathcal{Y}$, by calculating the most probable result of $f(\boldsymbol{x} + \epsilon)$ where $\epsilon \sim \mathcal{N}(0, \sigma^2 \mathbb{1})$:

$$g(\boldsymbol{x}) := \arg \max_c \mathbb{P}_{\epsilon \sim \mathcal{N}(0, \sigma^2 \mathbb{1})}(f(\boldsymbol{x} + \epsilon) = c).$$

One then obtains the following robustness guarantee:

**Theorem 3.1** (From [7]). *Suppose $c_A \in \mathcal{Y}$, $\underline{p_A}, \overline{p_B} \in [0, 1]$. If*

$$\mathbb{P}_{\epsilon}(f(\boldsymbol{x} + \epsilon) = c_A) \geq \underline{p_A} \geq \overline{p_B} \geq \max_{c \neq c_A} \mathbb{P}_{\epsilon}(f(\boldsymbol{x} + \epsilon) = c),$$

*then $g(\boldsymbol{x} + \delta) = c_A$ for all $\delta$ satisfying $\|\delta\|_2 \leq \frac{\sigma}{2}(\Phi^{-1}(\underline{p_A}) - \Phi^{-1}(\overline{p_B})) =: r_{\delta}$.*

We now generalize this theorem to parameterized transformations. Consider the composable transformations $\psi_{\beta} \colon \mathbb{R}^m \to \mathbb{R}^m$, satisfying $\psi_{\beta} \circ \psi_{\gamma} = \psi_{\beta + \gamma}$ for all $\beta, \gamma \in \mathbb{R}^d$. Then we can define a smoothed classifier $g \colon \mathbb{R}^m \to \mathcal{Y}$ analogously for a parametric transformation $\psi_{\beta}$ by

$$g(\boldsymbol{x}) = \arg \max_c \mathbb{P}_{\beta \sim \mathcal{N}(0, \sigma^2 \mathbb{1})}(f \circ \psi_{\beta}(\boldsymbol{x}) = c). \tag{1}$$

With that, we obtain the following robustness guarantee:

**Theorem 3.2.** *Let $\boldsymbol{x} \in \mathbb{R}^m$, $f \colon \mathbb{R}^m \to \mathcal{Y}$ be a classifier and $\psi_{\beta} \colon \mathbb{R}^m \to \mathbb{R}^m$ be a composable transformation as above. If*

$$\mathbb{P}_{\beta}(f \circ \psi_{\beta}(\boldsymbol{x}) = c_A) \geq \underline{p_A} \geq \overline{p_B} \geq \max_{c_B \neq c_A} \mathbb{P}_{\beta}(f \circ \psi_{\beta}(\boldsymbol{x}) = c_B),$$

*then $g \circ \psi_{\gamma}(\boldsymbol{x}) = c_A$ for all $\gamma$ satisfying $\|\gamma\|_2 \leq \frac{\sigma}{2}(\Phi^{-1}(\underline{p_A}) - \Phi^{-1}(\overline{p_B})) =: r_{\gamma}$. Further, if $g$ is evaluated on a proxy classifier $f'$ that behaves like $f$ with probability $1 - \rho$ and else returns an arbitrary answer, then $r_{\gamma} := \frac{\sigma}{2}(\Phi^{-1}(\underline{p_A} - \rho) - \Phi^{-1}(\overline{p_B} + \rho))$.*

The proof is similar to the one presented in Cohen et al. [7] and is given in App. A. The key difference is that we allow parameterized transformations $\psi$, while Cohen et al. [7] only allows additive noise.

# 4 Certification with interpolation and rounding errors

We now instantiate Theorem 3.2 for parameterized geometric image transformations $T_{\beta}$, $\beta \in \mathbb{R}^d$, followed by interpolation $I$, denoted as $T_{\beta}^I$. A geometric transformation $T_{\beta}$ is followed by an interpolation $I$ in order to express the result on the pixel grid. In general, even if $T_{\beta}$ composes, $T_{\beta}^I$ does not (see Fig. 1 in the case where $T_{\beta}$ is a rotation $R_{\beta}$ by an angle $\beta$). This prevents us from directly instantiating Theorem 3.2 with $\psi_{\beta} := T_{\beta}^I$.

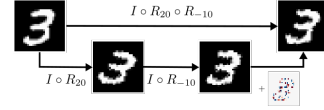

Figure 1: Rotations with interpolation do not compose.

To address this issue, we now show how to construct a classifier $g_E$ with the desired guarantees, namely that $g_E \circ T_{\gamma}^I(\boldsymbol{x}) = g_E(\boldsymbol{x})$ for $\gamma$ with $\|\gamma\|_2 \leq r_{\gamma}$, thus enabling certification of image transformations (which may not compose). Our proposed construction consists of two steps.

First, for a fixed but arbitrary $\boldsymbol{x}$, let $h_E$ be a classifier satisfying interpolation invariance:

$$h_E \circ T_{\beta}^I \circ T_{\gamma}^I(\boldsymbol{x}) = h_E \circ T_{\beta + \gamma}^I(\boldsymbol{x}) \quad \forall \beta, \gamma \in \mathbb{R}^d. \tag{2}$$

We now instantiate Theorem 3.2 with $f := h_E \circ I$ and $\psi_{\beta} := T_{\beta}$, obtaining a smoothed classifier $g_E(\boldsymbol{x}) := \arg \max_c \mathbb{P}_{\beta \sim \mathcal{N}(0, \sigma^2 \mathbb{1})}(h_E \circ I \circ T_{\beta}(\boldsymbol{x}) = c)$, such that $g_E \circ T_{\gamma}(\boldsymbol{x}) = c_A = g_E(\boldsymbol{x})$ for $\gamma$ with $\|\gamma\|_2 \leq r_{\gamma}$ by Theorem 3.2. Further, since

$$g_E \circ T_{\gamma}(\boldsymbol{x}) = \arg \max_c \mathbb{P}_{\beta \sim \mathcal{N}(0, \sigma^2 \mathbb{1})}(h_E \circ I \circ T_{\beta} \circ T_{\gamma}(\boldsymbol{x}) = c)$$

$$= \arg \max_c \mathbb{P}_{\beta \sim \mathcal{N}(0, \sigma^2 \mathbb{1})}(h_E \circ T_{\beta}^I \circ T_{\gamma}^I(\boldsymbol{x}) = c)$$

$$= g_E \circ T_{\gamma}^I(\boldsymbol{x}),$$

where the first and last equities hold by the definition of $g_E$ and the second one due to Eq. (2). Thus, we obtain a classifier $g_E$ with the desired property.

Second, we discuss the construction of the desired $h_E$ (from step 1). Consider the interpolation error

$$\epsilon(\beta, \gamma, \boldsymbol{x}) := T_\beta^I \circ T_\gamma^I(\boldsymbol{x}) - T_{\beta+\gamma}^I(\boldsymbol{x}), \tag{3}$$

$$\text{bounded by } E \in \mathbb{R}^{\geq 0} \text{ s.t. } \forall \beta, \gamma \in \mathbb{R}^d. \ \|\epsilon(\beta, \gamma, \boldsymbol{x})\|_2 \leq E \tag{4}$$

for a given but arbitrary $\boldsymbol{x}$. Thus if $h_E$ is $\ell^2$-robust with radius $E$ around $T_{\beta+\gamma}^I(\boldsymbol{x})$, interpolation invariance holds. While many choices for such $h_E$ are possible in the rest of the paper we instantiate $h_E$ by applying Theorem 3.1 to a base classifier $b$.

**Obtaining probabilistic guarantees from Theorem 3.2**  So far we assumed that $\boldsymbol{x}$ is arbitrary but fixed and constructed $E$ and $h_E$ for this $\boldsymbol{x}$ specifically. In general, finding a tight deterministic bound $E$ that holds $\forall \beta, \gamma$ is computationally challenging. Thus, we relax this deterministic guarantee into a probabilistic one:

$$\mathbb{P}_{\beta \sim \mathcal{N}(0,\sigma^2\mathbb{1})} \left( \|\epsilon(\beta, \gamma, \boldsymbol{x})\|_2 \leq E \right) \geq 1 - \alpha_E \ \ \forall \gamma \in \mathbb{R}^d. \tag{5}$$

meaning Eq. (4) holds with probability at least $1 - \alpha_E$, in turn implying that Eq. (2) also holds at least with probability $1 - \alpha_E$. This can also be formulated as having a proxy classifier $h_E'$ which behaves like $h_E$ with probability at least $1 - \alpha_E$ on the inputs specified by Eq. (2). In practice, we construct $h_E'$ which behaves like $h_E$ with probability at least $1 - \alpha_E$ on all inputs, implying this behavior on the inputs from Eq. (2). From $h_E'$, we then obtain $f' := h_E' \circ I$ which behaves like $f$ with probability at least $1 - \alpha_E$ on all inputs. Then, we can apply Theorem 3.2 by setting $\rho$ to $\alpha_E$ and obtain the desired guarantee. In Section 5, we show how to obtain $E$ for DISTSPT and INDIVSPT.

# 5 Calculation of error bounds

In Section 5.1 we derive a distributional error bound over a dataset and in Section 5.2 a per-image bound. Throughout this section, we assume the attacker model $\gamma \in \Gamma \subseteq \mathbb{R}^d$. As we compute $E$ with this assumption, our obtained certificate proves robustness of $g_E$ to $T_\gamma^I$ for $\gamma \in \Gamma$ with $\|\gamma\|_2 \leq r_\gamma$.

## 5.1 Distributional bounds for DISTSPT

For a fixed $E \in \mathbb{R}^{\geq 0}, \alpha_E \in [0, 1]$, the probability that $\epsilon$ is bounded by $E$ for $\boldsymbol{x} \sim \mathcal{D}$ is

$$q_E := \mathbb{P}_{\boldsymbol{x} \sim \mathcal{D}}(\mathbb{P}_{\beta \sim \mathcal{N}(0,\sigma^2\mathbb{1})}(\max_{\gamma \in \Gamma} \|\epsilon(\beta, \gamma, \boldsymbol{x})\|_2 \leq E) \geq 1 - \alpha_E). \tag{6}$$

In practice, we evaluate $q_E$ by sampling $\boldsymbol{x}$ and counting how often the inner property holds. We compute the inner probability by: (i) sampling multiple realizations of $\beta$, (ii) computing their corresponding error $\epsilon$ and checking how many are successfully bounded by $E$, and (iii) bounding the inner probability using Clopper-Pearson. If this lower bound is larger than $1 - \alpha_E$ we count this as a positive sample, else a negative. Once these counts are obtained for a number of sampled points $\boldsymbol{x}$, we can apply Clopper-Pearson and obtain a lower bound $\underline{q_E}$ with the desired confidence.

To compute the maximization over $\gamma$ we employ standard interval analysis, which allows us to efficiently propagate lower and upper bounds [27]. By propagating the hyperrectangle containing $\Gamma$ along with the sampled $\beta$ and $\boldsymbol{x}$, we eventually obtain a lower and upper bound for the norm calculation of which we take the maximum:

$$\max_{\gamma \in \Gamma} \|\epsilon(\beta, \gamma, \boldsymbol{x})\|_2 \leq \max \|T_\beta^I \circ T_\Gamma^I(\boldsymbol{x}) - T_{\beta+\Gamma}^I(\boldsymbol{x})\|_2. \tag{7}$$

The result can be refined by splitting the hyperrectangle $\Gamma$ into smaller hyperrectangles $\Gamma_k$ for $k \in \{0, \dots, N\}$. The refined bound is

$$\max_{\gamma \in \Gamma} \|\epsilon(\beta, \gamma, \boldsymbol{x})\|_2 \leq \max_{k \in \{0,\dots,N\}} \max \|T_\beta^I \circ T_{\Gamma_k}^I(\boldsymbol{x}) - T_{\beta+\Gamma_k}^I(\boldsymbol{x})\|_2. \tag{8}$$

To obtain $E$ in the first place, we perform the same sampling operations as above (sample $\boldsymbol{x}$ and $\beta$) but do not compute any probabilities, that is, for each sample $(\boldsymbol{x}, \beta)$, we simply keep the values attained by Eq. (8). We then pick an $E$ that bounds many of these values, choosing $\alpha_E$ to be small. Once $E$ is obtained, we compute $q_E$ as described above. Instantiating the construction of Section 4 with this $E$ yields the guarantee that for a random image $\boldsymbol{x} \sim \mathcal{D}$ the guarantees provided by Theorem 3.2 hold with probability $q_E$.

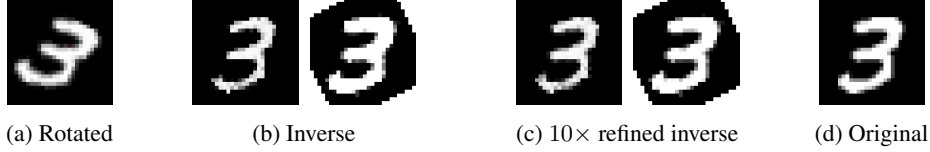

| (a) Rotated | (b) Inverse | (c) $10\times$ refined inverse | (d) Original |

Figure 2: Over approximation of the inverse image. The image pairs (b) and (c) depict the lower (left) and upper (right) interval pixel bounds for the inverse image and the $10\times$ refined image respectively.

## 5.2 Individual bounds for INDIVSPT

At inference time, we are given $\boldsymbol{x}' := T_\gamma^I(\boldsymbol{x})$ but neither the original $\boldsymbol{x}$ nor the parameter $\gamma \in \Gamma$, and we would like to certify that $g_E(\boldsymbol{x}') = g_E(\boldsymbol{x})$. When $\psi_\beta$ composes as required in Section 3, this can be certified by showing $g$ is robust with a sufficient radius $r_\gamma$. However, when $\psi_\beta$ does not compose, this can be accomplished by applying Theorem 3.2 to show $g_E(\boldsymbol{x})$ is robust with radius $r_\gamma$ that includes $\Gamma$. In turn, this requires a bound $E$ (see Eq. (5)) for $\boldsymbol{x}$ (rather than $\boldsymbol{x}'$):

$$\mathbb{P}_{\beta\sim\mathcal{N}(0,\sigma^2\mathbb{1})}\left(\max_{\gamma\in\Gamma}\|\epsilon(\beta,\gamma,\boldsymbol{x})\|_2 \leq E\right) \geq 1-\alpha_E. \tag{9}$$

Now, we would like to compute an upper bound on the max term *without* having access to $\boldsymbol{x}$. This is accomplished as follows. First, in the above equation, we replace $\epsilon$ by its definition (Eq. (3)) and $T_\gamma^I(\boldsymbol{x})$ by $\boldsymbol{x}'$. We then replace $\boldsymbol{x}$ with a symbolic set of possible inputs that could have generated $\boldsymbol{x}'$, denoted as $(T_\Gamma^I)^{-1}(\boldsymbol{x}') := \{\boldsymbol{x} \in \mathbb{R}^m \mid T_\gamma^I(\boldsymbol{x}) = \boldsymbol{x}', \gamma \in \Gamma\}$ which we can use instead of $\boldsymbol{x}$ due to the maximization over $\gamma$. As in Section 5.1, we obtain the resulting bound via interval analysis:

$$\max_{\gamma\in\Gamma}\|\epsilon(\beta,\gamma,\boldsymbol{x})\|_2 \leq \max\|T_\beta^I(\boldsymbol{x}') - T_{\beta+\Gamma}^I \circ (T_\Gamma^I)^{-1}(\boldsymbol{x}')\|_2. \tag{10}$$

The computation of the inverse $(T_\Gamma^I)^{-1}(\boldsymbol{x}')$ is explained in Section 6. By substituting Eq. (10) in Eq. (9) we can obtain and verify $E$ as in Section 5.1 (except we do not need to sample $\boldsymbol{x}$'s). As before, we can refine the upper bound of Eq. (10) by splitting $\Gamma$ into $\Gamma_k$. We note as the inverse does not depend on $\beta$, given $\boldsymbol{x}'$, it only needs to be computed once and can be reused whenever we evaluate Section 5.1 for a given sample $\beta$.

## 6 Inverse Computation

We now discuss how to obtain a set containing all possible inverse images. That is, given $\boldsymbol{x}' := T_\gamma^I(\boldsymbol{x})$ and $\gamma \in \Gamma$, we compute the set $(T_\Gamma^I)^{-1}(\boldsymbol{x}')$ which contains all possible $\boldsymbol{x}$. First, we cover the necessary background. To ease presentation, we assume even image height and width. We embed the images in $\mathbb{R}^2$ by centering them at 0 on an odd integer grid $G := (2\mathbb{Z}+1) \times (2\mathbb{Z}+1)$ and centered at 0. We denote the value of a pixel at $(i,j) \in G$ by $p_{i,j} \in [0,1]$.

**Transformations**   The pixel values $p'_{i',j'}$ for $(i',j') \in G$ of an image, produced by a transformation $T_\gamma\colon \mathbb{R}^2 \to \mathbb{R}^2$ with parameter $\gamma \in \mathbb{R}^d$, is calculated by interpolating at the inversely transformed coordinate $T_\gamma^{-1}(i',j')$, followed by the interpolation $I$ resulting in $p'_{i',j'} = I \circ T_\gamma^{-1}(i',j')$.

**Bilinear interpolation**   A prominent interpolation is *bilinear interpolation*, given by

$$I(x,y) = p_{v,w}\frac{2+v-x}{2}\frac{2+w-y}{2} + p_{v,w+2}\frac{2+v-x}{2}\frac{y-w}{2} + p_{v+2,w}\frac{x-v}{2}\frac{2+w-y}{2} + p_{v+2,w+2}\frac{x-v}{2}\frac{y-w}{2}, \tag{11}$$

where $(v,w) \in G$ is the coordinate such that $(x,y)$ lies in the $(v,w)$-interpolation region, that is $(x,y) \in [v,v+2] \times [w,w+2]$. We use $v$ and $w$ as grid indices in the context of the interpolation $I$. If $p_{v,w}$ has no defined value because $(v,w)$ is out of range for the image, we set $p_{v,w}$ to 0.

We start by giving a procedure to calculate constraints of a single pixel $(i,j)$ for a single color channel, after which we present an iterative procedure to refine that constraint. The inverse image is then obtained by following this procedure for every pixel in every color channel. We illustrate the steps in Section 6.1 using the example of a rotated image $\boldsymbol{x}'$ (Fig. 2a).

The attacker transformed the original image $\boldsymbol{x}$ (Fig. 2d) using $T_\gamma^I$ for $\gamma \in \Gamma$ and therefore obtained the pixel values $p'_{i',j'}$ of the transformed image $\boldsymbol{x}'$ by evaluating $p'_{i',j'} = I \circ T_\gamma^{-1}(i',j')$. The interpolation $I$ uses the pixel values $p_{i,j}$ of $\boldsymbol{x}$. The following steps invert this relation for every coordinate $(i,j)$:

**Step 1**  For every $(i', j') \in G$, we over-approximate the region the pixel value $p'_{i',j'}$ could have been interpolated from, which is $c_{i',j'} := T_\Gamma^{-1}(i', j')$, $C := \{c_{i',j'} \mid (i', j') \in G\}$. In practice, only a finite subset of $C$ is used. In App. B, we show how to calculate this subset efficiently.

**Step 2**  The interpolation $I$ is defined piecewise per $(v, w)$-interpolation region $[v, v+2] \times [w, w+2]$, so the algebraic form of $I$, Eq. (11) holds for each interpolation region separately. For every interpolation region cornering $(i, j)$ that $c_{i',j'}$ intersects with, the pixel value $p'_{i',j'}$ yields constraints for value $p_{i,j}$. Here, we describe just the constraint $q_{i,j}$ associated with the $(i, j)$-interpolation region; others $((i-2, j-2), (i-2, j), (i, j-2))$ work analogously. First, for every $c_{i',j'} \in C$ we calculate its intersection with the $(i, j)$-interpolation region, yielding

$$[x_l, x_u] \times [y_l, y_u] := c_{i',j'} \cap [i, i+2] \times [j, j+2].$$

We can plug this into the interpolation $I$, where we instantiate $(v, w) \leftarrow (i, j)$, resulting into

$$p'_{i',j'} \in I([x_l, x_u], [y_l, y_u]) = p_{i,j} \frac{2+i-[x_l,x_u]}{2} \frac{2+j-[y_l,y_u]}{2} + p_{i,j+2} \frac{2+i-[x_l,x_u]}{2} \frac{[y_l,y_u]-j}{2}$$
$$+ p_{i+2,j} \frac{[x_l,x_u]-i}{2} \frac{2+j-[y_l,y_u]}{2} + p_{i+2,j+2} \frac{[x_l,x_u]-i}{2} \frac{[y_l,y_u]-j}{2}. \tag{12}$$

Next, we solve for the pixel value of interest $p_{i,j}$. Then, we replace all other three pixel values $p_{i,j+2}$, $p_{i+2,j}$, and $p_{i+2,j+2}$ with the (trivial) $[0, 1]$ constraint, covering all possible pixel values. While this results into sound constraints for $p_{i,j}$, instantiating $[x_l, x_u]$ and $[y_l, y_u]$ with its corner $(x, y)$ furthest from $(i, j)$, yields still a sound but more precise constraint $q_{i,j}$ for $p_{i,j}$. Here, this amounts to $x \leftarrow x_u$ and $y \leftarrow y_u$. App. B presents a detailed explanation of the derivation. The result is

$$q_{i,j} = \left[ p'_{i',j'} - \left( \frac{2+i-x_u}{2} \frac{y_u-j}{2} + \frac{x_u-i}{2} \frac{2+j-y_u}{2} + \frac{x_u-i}{2} \frac{y_u-j}{2} \right), p'_{i',j'} \right] \left( \frac{2+i-x_u}{2} \frac{2+j-y_u}{2} \right)^{-1}.$$

**Step 3**  In order to be sound, we need to take the union over $q_{i-2,j-2}, q_{i-2,j}, q_{i,j-2}, q_{i,j}$ for each $c_{i',j'}$. To gain precision, we can intersect all of those unions and finally, we can intersect this constraint with the trivial one, $[0, 1]$, resulting in the final pixel constraint for pixel $p_{i,j}$:

$$p_{i,j} \in [0, 1] \cap \left( \bigcap_{c_{i',j'} \in C} q_{i-2,j-2}(c_{i',j'}) \sqcup q_{i,j-2}(c_{i',j'}) \sqcup q_{i-2,j}(c_{i',j'}) \sqcup q_{i,j}(c_{i',j'}) \right), \tag{13}$$

where $\sqcup$ denotes the *join* operation, that is $[a, b] \sqcup [c, d] := [\min(a, c), \max(b, d)]$. If the intersection of $c_{i',j'}$ with the respective $(v, w)$-interpolation region is empty, we omit $q_{v,w}$ in Eq. (13).

In Section 5.2, we split $\Gamma$ into $\Gamma_k$. It often happens that one of the resulting intervals is empty. Then we know for sure that $\gamma$ lies in a different $\Gamma_k$, speeding up the process substantially.

**Refined Inverse**  The constraints can be refined by following the same steps as for calculating the inverse, but instead of replacing the (unknown) pixel values in Eq. (12) with $[0, 1]$, we replace them with the intervals calculated previously. However, replacing $[x_l, x_u] \times [y_l, y_u]$ with the corner furthest away from $(i, j)$ would be unsound. To be sound, one needs to consider all 4 corners of every non-empty intersection $[x_l, x_u] \times [y_l, y_u]$ and join all interval constraints. Similarly, we use the previously calculated constraint for $p_{i,j}$ instead of $[0, 1]$ in Eq. (13). This procedure can be repeated to further increase precision. The final result after applying the refinement 10 times is shown in Fig. 2c representing the lower (left) and upper (right) interval bound for all pixels.

## 6.1  Example

We calculate the constraint for pixel $(3, 3)$ of the original image (Fig. 2d), depicted as the green dot in Fig. 3 under the assumption $\gamma \in [23°, 26°]$. We elaborate the constraints that pixel $(5, 1)$ of the rotated image (Fig. 2a) yields for pixel $(3, 3)$ of the original image.

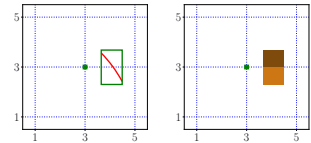

(a) $c_{5,1}$      (b) Intersections

**Step 1**  We illustrate the calculation of the set $C$ for $c_{5,1} := R_{[23°,26°]}^{-1} \binom{5}{1} = \binom{[4.06,4.21]}{[2.85,3.11]}$. The result is depicted as the green box in Fig. 3a enclosing the red arc. The red arc shows the precise set of coordinates where the pixel value $p'_{5,1}$ could have been interpolated from the original image $\boldsymbol{x}$.

Figure 3: To improve presentation, the red arc is $3\times$ longer.

**Step 2** The only non-empty intersections of $c_{5,1}$ with interpolation regions (blue squares in Fig. 3), cornering $(3,3)$ are the $(3,1)$ and the $(3,3)$-interpolation regions, hence we omit $q_{1,1}$ and $q_{1,3}$. The intersection with the $(3,3)$-interpolation region yields $[x_l, x_u] = [4.06, 4.21]$ and $[y_l, y_u] = [3, 3.11]$ (dark brown rectangle in Fig. 3b), hence at the furthest corner $(x,y) = (4.21, 3.11)$, we get

$$q_{3,3} = [0.73, 2.48] = \left[ p'_{5,1} - \left( \tfrac{5-x}{2}\tfrac{y-3}{2} + \tfrac{x-3}{2}\tfrac{5-y}{2} + \tfrac{x-3}{2}\tfrac{y-3}{2} \right), p'_{5,1} \right] \left( \tfrac{5-x}{2}\tfrac{5-y}{2} \right)^{-1},$$

and the intersection with the $(3,1)$-interpolation region yields $[x_l, x_u] = [4.06, 4.21]$ and $[y_l, y_u] = [2.85, 3]$ (light brown rectangle in Fig. 3b), hence at the furthest corner $(x,y) = (4.21, 2.85)$, we get

$$q_{3,1} = [0.72, 2.48] = \left[ p'_{5,1} - \left( \tfrac{5-x}{2}\tfrac{3-y}{2} + \tfrac{x-3}{2}\tfrac{3-y}{2} + \tfrac{x-3}{2}\tfrac{y-1}{2} \right), p'_{5,1} \right] \left( \tfrac{5-x}{2}\tfrac{y-1}{2} \right)^{-1}.$$

**Step 3** The join $q_{3,1} \sqcup q_{3,3}$ yields $[0.72, 2.48]$. After intersecting this with $[0,1]$ and the constraints from the other $c_{i',j'} \in C$ (as in Eq. (13)), we are left with the final result $p_{3,3} \in [0.73, 1]$.

The final result of the inverse calculation for all pixels is shown in Fig. 2b representing the lower (left) and upper (right) interval bounds for all pixels. The iterative refinement is shown in Fig. 2c.

# 7 Experimental Evaluation

We now present our extensive evaluation of the different defenses discussed so far.

## 7.1 Instantiation in Practice

In Section 4 we showed how to certify robustness of $g_E$ to $T_\gamma^I$, obtained from Eq. (1) with $f := h_E \circ I$ and $\psi_\beta := T_\beta$. Since in practice $h_E \circ I$ and $T_\beta$ cannot be evaluated as $I$ and $T_\beta$ are not available independently, in order to evaluate $g_E$ in practice, we need to re-write it as follows:

$$g_E(\boldsymbol{x}) = \arg\max_c \mathbb{P}_{\beta \sim \mathcal{N}(0,\sigma^2 \mathbb{1})} \left( (h_E \circ I) \circ T_\beta(\boldsymbol{x}) = c \right)$$
$$= \arg\max_c \mathbb{P}_{\beta \sim \mathcal{N}(0,\sigma^2 \mathbb{1})} \left( h_E \circ (I \circ T_\beta)(\boldsymbol{x}) = c \right) =: g(\boldsymbol{x}),$$

which is an instantiation of Eq. (1) with $f := h_E$ and $\psi_\beta := T_\beta^I$, both of which are available.

Further, as the probability in Eq. (1) cannot be computed exactly, in practice we use the approximation introduced in Cohen et al. [7]: by taking $n$ samples around a given $\boldsymbol{x}$ with standard deviation $\sigma$, we can obtain $g(\boldsymbol{x})$ and the corresponding robustness radius $r$ with confidence $1 - \alpha$. Here, $n$ can be too small to make a statement with confidence $1 - \alpha$, in which case the classifier abstains. Further, we let $\sigma_\gamma, \alpha_\gamma, n_\gamma, r_\gamma$ and $\sigma_\delta, \alpha_\delta, n_\delta, r_\delta$ denote the parameters and radius required to use Theorem 3.2 and Theorem 3.1 in practice, respectively. As we require $h_E$ to be robust with radius at least $E$, we treat abstentions and certification with $r_\delta < E$ as incorrect classifications. As the certificate of $h_E$ can be inaccurate with probability $\alpha_\delta$, we need to further increase $\rho = \alpha_E$ (from Section 4) to $\rho = \alpha_\delta + \alpha_E$.

## 7.2 Setup

All experiments were performed on a machine with 2 GeForce RTX 2080 Tis and an Intel(R) Core(TM) i9-9900K CPU. As base classifiers $b$ we utilize neural networks in PyTorch [28], using `robustness` [29] for training. Further, we implemented the interval analysis (cf. §5 and §6) of the interpolation error and inverse computation in C++/CUDA.

In our evaluation we consider rotations $R_\gamma^I$ by $\gamma$ degrees and translations $\Delta_\gamma^I$ by $\gamma \in \mathbb{R}^2$ with bilinear interpolation $I$. Here, we allow the adversary to choose $\gamma \in \Gamma$. For a scalar $\Gamma_\pm \in \mathbb{R}^{\geq 0}$, we permit $\Gamma := [-\Gamma_\pm, \Gamma_\pm]$ for rotations and $\Gamma := [-\Gamma_\pm, \Gamma_\pm]^2$ for translations. All estimates of $E$ include interpolation errors as well as 8-bit representation ("rounding") errors.

We evaluate on ImageNet [30], Restricted ImageNet (RImageNet)[31], a subset of ImageNet with 10 classes, CIFAR-10 [32], and MNIST [33]. For the base classifier, in Section 7.3 we use standard models without any additional training, while in the other sections we use models trained with data augmentation (transformations, $\ell^2$-noise) using [8].

In §7.4 and §7.5, we apply a circular or rectangular vignette for rotation and translation respectively, to reduce error estimates in areas of the image where information is lost. We also apply a Gaussian blur prior to classification to further reduce the high-frequency components of the interpolation error. App. D contains further details on prepossessing, model training and parameters. Note that pre-processing does not impact the theoretical guarantees as long as it is consistently applied. We provide an ablation study regarding vignetting and Gaussian blur in App. F. Additional experiments, including other interpolation methods or audio classification are provided in App. E, highlighting the generality of our methods. Throughout the section we use $\alpha_\delta = 0.002$ and $\alpha_\gamma = 0.01$ for confidences.

Table 1: Evaluation of BASESPT. We obtain Acc for $b$ on the test set and evaluate adv. Acc. on 3000 images obtained by the `worst-of-100` attack. $t$ denotes the average run time of $g$.

| Dataset | $T^I$ | $\Gamma_\pm$ | Acc. $b$ | adv. Acc. $b$ | $g$ | t [s] |
|---|---|---|---|---|---|---|
| MNIST | $R^I$ | 30° | 0.99 | 0.73 | 0.99 | 0.97 |
| CIFAR-10 | $R^I$ | 30° | 0.91 | 0.26 | 0.85 | 0.95 |
| ImageNet | $R^I$ | 30° | 0.76 | 0.56 | 0.76 | 5.43 |
| MNIST | $\Delta^I$ | 4 | 0.99 | 0.03 | 0.53 | 0.86 |
| CIFAR-10 | $\Delta^I$ | 4 | 0.91 | 0.44 | 0.79 | 0.95 |
| ImageNet | $\Delta^I$ | 20 | 0.76 | 0.65 | 0.75 | 6.70 |

## 7.3 BASESPT

We can quickly obtain a well-motivated but empirical defense by instantiating Theorem 3.2 with $\psi_\beta := T^I_\beta$ and ignoring both the interpolation error Eq. (3) and the construction in Section 4. Table 1 shows results on an undefended classifier $b$ and the BASESPT smoothed version $g$. Here *Acc.* is obtained over the whole dataset. To evaluate *adv. Acc.* we use the `worst-of-k` proposed by Engstrom et al. [2], which returns the $\gamma$ yielding the highest cross-entropy loss out of $k$ randomly sampled $\gamma \sim \mathcal{U}(\Gamma)$. We apply `worst-of-k` to 1000 images and produce 3 attacked images each, resulting 3000 samples on which we then evaluate $b$ and $g$. For $g$, the average inference time per image $t$ is generally fast, where most time is spent on sampling transformations. The actual inference, invoking $b$ on the samples, is not slowed down as all samples fit into a single batch. In this section we use $n_\gamma = 1000, \sigma_\gamma = \Gamma_\pm$. We do not obtain certificates here as the assumptions of Theorem 3.2 are violated. However, we investigate in App. E if the certification radius holds practically.

## 7.4 DISTSPT

In Table 2, we show our results for DISTSPT with rotations. We restrict the attacker model to $\Gamma_\pm = 30°$. To obtain $E$, we used 1000 images from the training set for MNIST and CIFAR-10 and 700000 for ImageNet to compute the bounded interpolation error, Eq. (7). The largest of these is shown as $\epsilon_{\max}$. We choose $E$ roughly 1.5 times larger. For small images this bound can be computed quickly. However for large images (ImageNet), the optimization over $\gamma$ for many images is computationally expensive. Thus, for ImageNet we replace the $\max$ in Eq. (6) with the maximum over 10 samples $\gamma \in \mathcal{U}(\Gamma)$. This formally restricts the certificate to only hold against random attacks. However, if sufficient computational resources are available, the $\max$ method can still be applied (we empirically find the method to obtain similar values). On (R)ImageNet (variable image size) we resize all images so that the short side is 512 pixel prior to applying transformations. As RImageNet is a subset of ImageNet we use the same bound. Here we use $\alpha_E = 0.001$ and expect $q_E$ to be close to 1 for all datasets. We show $q_E \geq 0.99$ with confidence 0.999 by using 1000 samples for $x$ and 8000 for $\beta$ (and correction for possible test errors over $\beta$). Subsequently, we evaluate the accuracy of $b$ and $g$. For $b$ we use the whole test set, while for $g$ we use 1000 samples. We clip obtained robustness radius $r_\gamma$ to $\Gamma$ (indicated by $^\dagger$) in order to provide a sound guarantee. Additionally, we evaluate rotations on MNIST with $\Gamma_\pm = 180°$, where we also obtain $E = 0.55$ (thus not further restricting the 30° results) and show robustness for 0.84 of images with a median $r_\gamma$ of $180^\dagger$. Finally, we evaluate translations on MNIST ($E = 0.72, \Gamma_\pm = 2$) and achieve 0.95 certified accuracy with $r_\gamma$ of 2.41 and cover all of $\Gamma$ for the $25^{\text{th}}$ and $50^{\text{th}}$ percentile respectively. The results on RImageNet indicate that the limiting factor for our method is the robustness and quality of the base classifier, not the size of the image. $\sigma_\delta = 0.3$ for MNIST, $\sigma_\delta = 0.25$ for CIFAR-10 and $\sigma_\delta = 0.5$ for (R)ImageNet.

**Comparison to other work** Balunovic et al. [11] certifies model accuracy on the test set and thus provides a distributional bound. On MNIST they report $87.01\%$ of certified accuracy for rotations with $\pm 30°$ ($35s$ per image), which with further refinement (at cost of run time) can be increased to $97\%$, and for translations with $\pm 2$ pixels $76.30\%$ (263 s per image). We significantly improve upon

Table 2: Evaluation of DISTSPT for $T^I := R^I$. $\epsilon_{\max}$ is computed on the training set. We show the test set accuracy of $b$, certified accuracy of $g$ and distribution of the obtained certification radius $r_\gamma$, along with the average run time $t$ and the number of used samples $n_\gamma, n_\delta$.

| Dataset | $\epsilon_{\max}$ | $E$ | Acc. | | $r_\gamma$ percentile | | | t [s] | $n_\gamma$ | $n_\delta$ |
| | | | $b$ | $g$ | $25^{\text{th}}$ | $50^{\text{th}}$ | $75^{\text{th}}$ | | | |
|---|---|---|---|---|---|---|---|---|---|---|
| MNIST | 0.36 | 0.55 | 0.99 | 0.97 | 38.24 | 44.90 | 54.07 | 2.55 | 200 | 200 |
| CIFAR-10 | 0.51 | 0.77 | 0.72 | 0.54 | 1.87 | 18.37 | $30.00^\dagger$ | 23.93 | 50 | 10000 |
| CIFAR-10 | 0.51 | 0.77 | 0.72 | 0.56 | 6.79 | 25.44 | $30.00^\dagger$ | 96.45 | 200 | 10000 |
| RImageNet | 0.91 | 1.20 | 0.84 | 0.73 | 12.15 | $30.00^\dagger$ | $30.00^\dagger$ | 85.58 | 50 | 2000 |
| ImageNet | 0.91 | 1.20 | 0.32 | 0.23 | 1.86 | 16.17 | 26.31 | 85.58 | 50 | 2000 |

this. On CIFAR-10 they certify rotation up to $10°$ for $62.51\%$, but unlike our work does not scale to larger image sizes and models, such as a ResNet-50 on ImageNet. We provide further comparison with Balunovic et al. [11] in App. F. Results, similar to ours, are obtained in [13]. However, they require access to the unperturbed image to evaluate their classifier, which is practically not feasible. Pei et al. [9] certify $\pm 2°$ in 714s per image on ImageNet. However, they report the failure rate per image and use nearest-neighbor interpolation and are thus incomparable.

### 7.5 INDIVSPT

Finally, we evaluate our certified online defense: where we compute $E$ on the given input. The bound computed by interval analysis is always sound, but may be quite large due to the loss of precision inherent in interval analysis. We show results for MNIST and discuss challenges on larger datasets in App. C. To this end, we attack images as in Section 7.3, and subsequently apply INDIVSPT. We use the `worst-of-100` attack on a base classifier $b$ to obtain a set of attacked images. To these images we then apply INDIVSPT. For rotations ($\Gamma_\pm = 10, \sigma_\gamma = 30, n_\gamma = 2000$, 3 attacks per image, 1000 images) we fix $E = 0.7$ and use 100 samples of $\beta$ to obtain the correct $\alpha_E$ (Eq. (9)). $g$ was correct on $98\%$ of attacked images. For $76\%$ of these, we could certify that the attacked image that classifies the same as the original. The analysis of $E$ took on average $0.95$s and the randomized smoothing $44.81$s. For translation ($\Gamma_\pm = 1, \sigma_\gamma = 1.5, \sigma_\delta = 0.3, n_\gamma = 200$, 3 attacks per image, 100 images) we started with $E = 0.35$. $g$ classified $71\%$ of attacked images correctly and could certify $88\%$ of these while on average taking $58.40$s for analysis and $100.47$s for smoothing per image. The reason for the higher run time is that compared to rotation less possible inverses can be discarded. We use 10 refinement steps and $n_\delta = 200$ for both rotations and translations. In both settings the certification rate can be increased further, while also increasing evaluation time, by increasing $n_\gamma, \sigma_\gamma, n_\delta, \sigma_\delta$.

**Limitations & Generalization** While we showcased translation and rotation, our approach is not limited to these transformations or to specific interpolation methods. BASESPT and DISTSPT can be directly adapted to other transformations, interpolation schemes or domains such as audio (see App. E). INDIVSPT can also be adapted but requires additional care. Generally, Theorem 3.2 can be applied to all parameterized data transformations that are additive in the parameter space. If this holds up to a small error, as discussed here, DISTSPT and INDIVSPT can be applied. While many data transformations, e.g., image scaling are additive in their parameter space, their compositions are often not (e.g., rotation and translation). As we are most limited by the $\ell^p$-robustness of $b$, any gains in $\ell^p$ certification will directly improve our method. Further, INDIVSPT can incur a large loss of precision in the inverse computation. Improving this directly increases the applicability of the method.

## 8 Conclusion

We presented a generalization of randomized smoothing to image transformations, a challenging task as image transformations do not compose. Based on this generalization, we presented two certified defenses with distributional and individual guarantees (which relies on efficient inverse computation). Finally, we showed both defenses are applicable in an online setting and realistic datasets.

## 9 Broader Impact

In general, methods from artificial intelligence can be applied in beneficial and malicious ways. While this poses a threat in itself, verification techniques provide formal guarantees for the robustness of the model, independently of the intended use case. Certification techniques could therefore distinguish a potentially unstable model from a stable one in safety critical settings, e.g., autonomous driving. However, especially for regulators, it is of utter importance to understand the certified properties of different certification methods precisely, as to avoid legal model deployment in safety critical applications based on misconceptions.

## Acknowledgments and Disclosure of Funding

We thank all reviewers for their helpful comments and feedback. We do not have any additional funding or compensation to disclose.

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
