[Supplementary Material]

# Supplementary Material for
# Certified Defense to Image Transformations via Randomized Smoothing

## A   Proof of Theorem 3.2

We now proceed to proof Theorem 3.2. We achieve this by first proofing an auxiliary Theorem and Lemma, and then instantiating as a special case Theorem 3.2 of these slightly more general results.

**Theorem A.1.** *Let $\boldsymbol{x} \in \mathbb{R}^n$, $f : \mathbb{R}^m \to \mathcal{Y}$ be a classifier, $\psi_\beta : \mathbb{R}^n \to \mathbb{R}^m$ a composable transformation for $\beta \sim \mathcal{N}(0, \Sigma)$ with a symmetric, positive-definite covariance matrix $\Sigma \in \mathbb{R}^{m \times m}$. If*

$$\mathbb{P}_\beta(f \circ \psi_\beta(x) = c_A) = p_A \geq \underline{p_A} \geq \overline{p_B} \geq p_B = \max_{c_B \neq c_A} \mathbb{P}_\beta(f \circ \psi_\beta(x) = c_B),$$

*then $g \circ \psi_\gamma(\boldsymbol{x}) = c_A$ for all $\gamma$ satisfying*

$$\sqrt{\gamma^T \Sigma^{-1} \gamma} < \tfrac{1}{2}(\Phi^{-1}(\underline{p_A}) - \Phi^{-1}(\overline{p_B})) =: r_\gamma.$$

*Proof.* The assumption is

$$\mathbb{P}\left((f \circ \psi_\beta)(\boldsymbol{x}) = c_A\right) = p_A \geq \underline{p_A} \geq \overline{p_B} \geq p_B = \mathbb{P}\left((f \circ \psi_\beta)(\boldsymbol{x}) = c_B\right).$$

By the definition of $g$ we need to show that

$$\mathbb{P}\left((f \circ \psi_{\beta+\gamma})(\boldsymbol{x}) = c_A\right) \geq \mathbb{P}\left((f \circ \psi_{\beta+\gamma})(\boldsymbol{x}) = c_B\right).$$

We define the set $A := \{\boldsymbol{z} \mid \gamma^T \Sigma^{-1} \boldsymbol{z} \leq \sqrt{\gamma^T \Sigma^{-1} \gamma} \Phi(\underline{p_A})\}$. We claim that for $\beta \sim \mathcal{N}(0, \Sigma)$, we have

$$\mathbb{P}(\beta \in A) = \underline{p_A} \tag{14}$$
$$\mathbb{P}(f \circ \psi_{\beta+\gamma}(x) = c_A) \geq \mathbb{P}(\beta + \gamma \in A). \tag{15}$$

First, we show that Eq. (14) holds.

$$\begin{aligned}
\mathbb{P}(\beta \in A) &= \mathbb{P}(\gamma^T \Sigma^{-1} \beta \leq \sqrt{\gamma^T \Sigma^{-1} \gamma} \Phi(\underline{p_A})) \\
&= \mathbb{P}(\gamma^T \Sigma^{-1} \mathcal{N}(0, \Sigma) \leq \sqrt{\gamma^T \Sigma^{-1} \gamma} \Phi(\underline{p_A})) \\
&= \mathbb{P}(\gamma^T \sqrt{\Sigma^{-1}} \mathcal{N}(0, \mathbb{1}) \leq \sqrt{\gamma^T \Sigma^{-1} \gamma} \Phi(\underline{p_A})) \\
&= \mathbb{P}(\mathcal{N}(0, \gamma^T \Sigma^{-1} \gamma) \leq \sqrt{\gamma^T \Sigma^{-1} \gamma} \Phi(\underline{p_A})) \\
&= \mathbb{P}(\sqrt{\gamma^T \Sigma^{-1} \gamma} \mathcal{N}(0, 1) \leq \sqrt{\gamma^T \Sigma^{-1} \gamma} \Phi(\underline{p_A})) \\
&= \mathbb{P}(\mathcal{N}(0, 1) \leq \Phi(\underline{p_A})) \\
&= \Phi(\Phi^{-1}(\underline{p_A})) \\
&= \underline{p_A}
\end{aligned}$$

Thus Eq. (14) holds. Next we show that Eq. (15) holds. For a random variable $v \sim \mathcal{N}(\mu_v, \Sigma_v)$ we write $p_v(z)$ for the evaluation of the Gaussian cdf at point $z$.

$$\mathbb{P}(f \circ \psi_{\beta+\gamma}(x) = c_A) - \mathbb{P}(\beta + \gamma \in A)$$

$$= \int_{\mathbb{R}^d} [f \circ \psi_{\boldsymbol{z}} = c_A] \, p_{\beta+\gamma}(z)dz - \int_A p_{\beta+\gamma}(z)dz$$

$$= \int_{\mathbb{R}^d \setminus A} [f \circ \psi_{\boldsymbol{z}}(x) = c_A] \, p_{\beta+\gamma}(z)dz + \int_A [f \circ \psi_{\boldsymbol{z}}(x) = c_A] \, p_{\beta+\gamma}(z)dz - \int_A p_{\beta+\gamma}(z)dz$$

$$= \int_{\mathbb{R}^d \setminus A} [f \circ \psi_{\boldsymbol{z}}(x) = c_A] \, p_{\beta+\gamma}(z)dz + \int_A [f \circ \psi_{\boldsymbol{z}}(x) = c_A] \, p_{\beta+\gamma}(z)dz$$

$$\quad - \left( \int_A [f \circ \psi_{\boldsymbol{z}}(x) = c_A] \, p_{\beta+\gamma}(z)dz + \int_A [f \circ \psi_{\boldsymbol{z}}(x) \neq c_A] \, p_{\beta+\gamma}(z)dz \right)$$

$$= \int_{\mathbb{R}^d \setminus A} [f \circ \psi_{\boldsymbol{z}}(x) = c_A] \, p_{\beta+\gamma}(z)dz - \int_A [f \circ \psi_{\boldsymbol{z}}(x) \neq c_A] \, p_{\beta+\gamma}(z)dz$$

$$\overset{Lemma\ 1}{\geq} t \left( \int_{\mathbb{R}^d \setminus A} [f \circ \psi_{\boldsymbol{z}}(x) = c_A] \, p_{\beta}(z)dz - \int_A [f \circ \psi_{\boldsymbol{z}}(x) \neq c_A] \, p_{\beta}(z)dz \right)$$

$$= t \left( \int_{\mathbb{R}^d} [f \circ \psi_{\boldsymbol{z}}(x) = c_A] \, p_{\beta}(z)dz - \int_A p_{\beta}(z)dz \right)$$

$$\overset{Eq.\ (14)}{\geq} 0.$$

Thus also Eq. (15) holds.

Next, we claim that for $B := \{z \mid \gamma^T \Sigma^{-1} \boldsymbol{z} \geq \sqrt{\gamma^T \Sigma^{-1} \gamma} \Phi^{-1}(1 - \overline{p_B})\}$ holds that

$$\mathbb{P}(f \circ \psi_\beta(x) = c_B) \leq \mathbb{P}(\beta \in B) \tag{16}$$

$$\mathbb{P}(f \circ \psi_{\beta+\gamma}(x) = c_B) \leq \mathbb{P}(\beta + \gamma \in B) \tag{17}$$

The proofs for Eq. (16) and Eq. (17) are analogous to the proofs for Eq. (14) and Eq. (15).

Now we derive the conditions that lead to $\mathbb{P}(\beta + \gamma \in A) > \mathbb{P}(\beta + \gamma \in B)$:

$$\mathbb{P}(\beta + \gamma \in A) = \mathbb{P}\left( \gamma^T \Sigma^{-1}(\beta + \gamma) \leq \sqrt{\gamma^T \Sigma^{-1} \gamma} \Phi^{-1}(\underline{p_A}) \right)$$

$$= \mathbb{P}\left( \gamma^T \Sigma^{-1}(\Sigma^{\frac{1}{2}} \mathcal{N}(0, \mathbb{1}) + \gamma) \leq \sqrt{\gamma^T \Sigma^{-1} \gamma} \Phi^{-1}(\underline{p_A}) \right)$$

$$= \mathbb{P}\left( \gamma^T \sqrt{\Sigma^{-1}} \mathcal{N}(0, \mathbb{1}) + \gamma^T \Sigma^{-1} \gamma \leq \sqrt{\gamma^T \Sigma^{-1} \gamma} \Phi^{-1}(\underline{p_A}) \right)$$

$$= \mathbb{P}\left( \sqrt{\gamma^T \Sigma^{-1} \gamma} \mathcal{N}(0, \mathbb{1}) + \gamma^T \Sigma^{-1} \gamma \leq \sqrt{\gamma^T \Sigma^{-1} \gamma} \Phi^{-1}(\underline{p_A}) \right)$$

$$= \mathbb{P}\left( \mathcal{N}(0, \mathbb{1}) + \sqrt{\gamma^T \Sigma^{-1} \gamma} \leq \Phi^{-1}(\underline{p_A}) \right)$$

$$= \mathbb{P}\left( \mathcal{N}(0, \mathbb{1}) \leq \Phi^{-1}(\underline{p_A}) - \sqrt{\gamma^T \Sigma^{-1} \gamma} \right)$$

$$= \Phi(\Phi^{-1}(\underline{p_A}) - \sqrt{\gamma^T \Sigma^{-1} \gamma})$$

Similarly, we have

$$\mathbb{P}(\beta + \gamma \in B) = \mathbb{P}\left( \mathcal{N}(0, \mathbb{1}) \geq \Phi^{-1}(1 - \overline{p_B}) - \sqrt{\gamma^T \Sigma^{-1} \gamma} \right)$$

$$= \Phi(\sqrt{\gamma^T \Sigma^{-1} \gamma} - \Phi^{-1}(1 - \overline{p_B}))$$

Thus, we get

$$\mathbb{P}(\beta + \gamma \in A) \qquad > \mathbb{P}(\beta + \gamma \in B)$$

$$\Leftrightarrow \quad \Phi(\Phi^{-1}(\underline{p_A}) - \sqrt{\gamma^T \Sigma^{-1} \gamma}) > \Phi(\sqrt{\gamma^T \Sigma^{-1} \gamma} - \Phi^{-1}(1 - \overline{p_B}))$$

$$\Leftrightarrow \quad \Phi^{-1}(\underline{p_A}) - \sqrt{\gamma^T \Sigma^{-1} \gamma} \qquad > \sqrt{\gamma^T \Sigma^{-1} \gamma} - \Phi^{-1}(1 - \overline{p_B})$$

$$\Leftrightarrow \quad \Phi^{-1}(\underline{p_A}) + \Phi^{-1}(1 - \overline{p_B}) > 2\sqrt{\gamma^T \Sigma^{-1} \gamma}$$

$$\Leftrightarrow \quad \tfrac{1}{2}(\Phi^{-1}(\underline{p_A}) - \Phi^{-1}(\overline{p_B})) \qquad > \sqrt{\gamma^T \Sigma^{-1} \gamma}.$$

$\square$

Next ,we show the lemma used in the proof.

**Lemma 1.** *There exists $t > 0$ such that $p_{\beta+\gamma}(z) \leq p_\beta(z) \cdot t$ for all $z \in A$. And further $p_{\beta+\gamma}(z) > p_\beta(z) \cdot t$ for all $z \in \mathbb{R}^d \setminus A$.*

*Proof.*

$$\frac{p_{\beta+\gamma}(z)}{p_\beta(z)} = \exp\left(-\tfrac{1}{2}(z-\gamma)^T\Sigma^{-1}(z-\gamma) + \tfrac{1}{2}z^T\Sigma^{-1}z\right)$$

$$= \exp\left(-\tfrac{1}{2}z^T\Sigma^{-1}z + z^T\Sigma^{-1}\gamma - \tfrac{1}{2}\gamma^T\Sigma^{-1}\gamma + \tfrac{1}{2}z^T\Sigma^{-1}z\right)$$

$$= \exp\left(z^T\Sigma^{-1}\gamma - \tfrac{1}{2}\gamma^T\Sigma^{-1}\gamma\right)$$

What is the lowest $t$ if it exists such that $\frac{p_{\beta+\gamma}(z)}{p_\beta(z)} \leq t$?

$$\frac{p_{\beta+\gamma}(z)}{p_\beta(z)} \leq t$$

$$\Leftrightarrow \quad \exp\left(z^T\Sigma^{-1}\gamma - \tfrac{1}{2}\gamma^T\Sigma^{-1}\gamma\right) \leq t$$

$$\Leftrightarrow \quad z^T\Sigma^{-1}\gamma - \tfrac{1}{2}\gamma^T\Sigma^{-1}\gamma \leq \log t$$

$$\Leftrightarrow \quad z^T\Sigma^{-1}\gamma \leq \log t + \tfrac{1}{2}\gamma^T\Sigma^{-1}\gamma$$

Because $z \in A$, we know that

$$z^T\Sigma^{-1}\gamma \leq \sqrt{\gamma^T\Sigma^{-1}\gamma}\,\Phi^{-1}(\underline{p_A}).$$

Does there exist a $t$ such that both upper bound coincide? Yes, namely

$$t = \exp\left(\sqrt{\gamma^T\Sigma^{-1}\gamma}\,\Phi^{-1}(\underline{p_A}) - \tfrac{1}{2}\gamma^T\Sigma^{-1}\gamma\right).$$

The case $p_{\beta+\gamma}(z) > p_\beta(z) \cdot t$ is analogous. $\square$

**Lemma 2.** *If we evaluate on a proxy classifier $f'$ instead of $f$, behaving with probability $(1-\rho)$ the same as $f$ and with probability $\rho$ differently than $f$ and if*

$$\mathbb{P}_{\beta,f'}(f' \circ \psi_\beta(x) = c_A) \geq \underline{p'_A} \geq \overline{p'_B} \geq \max_{c_B \neq c_A} \mathbb{P}_{\beta,f'}(f' \circ \psi_\beta(x) = c_B),$$

*then $g \circ \psi_\gamma(x) = c_A$ for all $\gamma$ satisfying*

$$\|\gamma\|_2 < \frac{\sigma}{2}(\Phi^{-1}(\underline{p'_A} - \rho) - \Phi^{-1}(\overline{p'_B} + \rho)).$$

*Proof.* By applying the union bound we can relate the output probability $p$ of $f$ for a class $c$ with the output probability of $f'$ and $p'$:

$$p' := \mathbb{P}_{\beta,f'}(f' \circ \psi_\beta(x) = c)$$

$$= \mathbb{P}_{\beta,f'}((f \circ \psi_\beta(x) = c) \vee (f' \text{ error}))$$

$$\leq \mathbb{P}_\beta(f \circ \psi_\beta(x) = c) + \mathbb{P}_{f'}(f' \text{ error})$$

$$= p + \rho$$

Thus we can obtain new bounds $\underline{p_A} \geq \underline{p'_A} - \rho$ and $\overline{p_B} \leq \overline{p'_B} + \rho$ from $\underline{p'_A}$ and $\overline{p'_B}$ measured on $f'$. Plugging these bounds in Theorem 3.2 yields the result. $\square$

We now show Theorem 3.2 (restarted below): Setting $\Sigma = \sigma^2 \mathbb{1}$ in Theorem A.1 directly recovers Theorem 3.2 up to the last sentence, which in turn is a direct consequence of Lemma 2.

**Theorem** (Theorem 3.2 restated). *Let $x \in \mathbb{R}^m$, $f : \mathbb{R}^m \to \mathcal{Y}$ be a classifier and $\psi_\beta : \mathbb{R}^m \to \mathbb{R}^m$ be a composable transformation as above. If*

$$\mathbb{P}_\beta(f \circ \psi_\beta(x) = c_A) \geq \underline{p_A} \geq \overline{p_B} \geq \max_{c_B \neq c_A} \mathbb{P}_\beta(f \circ \psi_\beta(x) = c_B),$$

*then $g \circ \psi_\gamma(x) = c_A$ for all $\gamma$ satisfying $\|\gamma\|_2 \leq \frac{\sigma}{2}(\Phi^{-1}(\underline{p_A}) - \Phi^{-1}(\overline{p_B})) =: r_\gamma$. Further, if $g$ is evaluated on a proxy classifier $f'$ that behaves like $f$ with probability $1 - \rho$ and else returns an arbitrary answer, then $r_\gamma := \frac{\sigma}{2}(\Phi^{-1}(\underline{p_A} - \rho) - \Phi^{-1}(\overline{p_B} + \rho))$.*

# B  Inverse and Refinement

## B.1  Details for Step 2

In this section, we elaborate on the details of Step 2 in Section 6. We consider the intersection of $c_{i',j'}$ with the $(i,j)$-interpolation region, $[x_l, x_u] \times [y_l, y_u] := c_{i',j'} \cap [i, i+2] \times [j, j+2]$. This yields,

$$p'_{i',j'} \in I([x_l, x_u], [y_l, y_u]) = p_{i,j} \frac{2+i-[x_l,x_u]}{2} \frac{2+j-[y_l,y_u]}{2} + p_{i,j+2} \frac{2+i-[x_l,x_u]}{2} \frac{[y_l,y_u]-j}{2}$$
$$+ p_{i+2,j} \frac{[x_l,x_u]-i}{2} \frac{2+j-[y_l,y_u]}{2} + p_{i+2,j+2} \frac{[x_l,x_u]-i}{2} \frac{[y_l,y_u]-j}{2}.$$

Next, we solve for the pixel value $p_{i,j}$ to get the constraint $q_{i,j}$:

$$q_{i,j} = \left( p'_{i',j'} - p_{i,j+2} \frac{2+i-[x_l,x_u]}{2} \frac{[y_l,y_u]-j}{2} - p_{i+2,j} \frac{[x_l,x_u]-i}{2} \frac{2+j-[y_l,y_u]}{2} \right.$$
$$\left. - p_{i+2,j+2} \frac{[x_l,x_u]-i}{2} \frac{[y_l,y_u]-j}{2} \right) \left( \frac{2+i-[x_l,x_u]}{2} \frac{2+j-[y_l,y_u]}{2} \right)^{-1}$$

Because we don't have any constraints for the pixel values $p_{i+2,j}, p_{i,j+2}$ and $p_{i+2,j+2}$, we replace their values by the $[0,1]$ constraint and obtain:

$$q_{i,j} = \left( p'_{i',j'} - \left( \frac{2+i-[x_l,x_u]}{2} \frac{[y_l,y_u]-j}{2} - \frac{[x_l,x_u]-i}{2} \frac{2+j-[y_l,y_u]}{2} \right.\right.$$
$$\left.\left. - \frac{[x_l,x_u]-i}{2} \frac{[y_l,y_u]-j}{2} \right) [0,1] \right) \left( \frac{2+i-[x_l,x_u]}{2} \frac{2+j-[y_l,y_u]}{2} \right)^{-1}$$

Instead of using standard interval analysis to compute the constraints for $p_{i,j}$, we use the following more efficient transformer: We replace $[x_l, x_u]$ and $[y_l, y_u]$ with the coordinate $(x, y) \in [x_l, x_u] \times [y_l, y_u]$ furthest away from $(i,j)$, which is in our case $(x_u, y_u)$ to obtain

$$q_{i,j} = \left( p'_{i',j'} - \left( \frac{2+i-x_u}{2} \frac{y_u-j}{2} + \frac{x_u-i}{2} \frac{2+j-y_u}{2} + \frac{x_u-i}{2} \frac{y_u-j}{2} \right) [0,1] \right) \left( \frac{2+i-x_u}{2} \frac{2+j-y_u}{2} \right)^{-1}$$
$$= \left[ p'_{i',j'} - \left( \frac{2+i-x_u}{2} \frac{y_u-j}{2} + \frac{x_u-i}{2} \frac{2+j-y_u}{2} + \frac{x_u-i}{2} \frac{y_u-j}{2} \right), p'_{i',j'} \right] \left( \frac{2+i-x_u}{2} \frac{2+j-y_u}{2} \right)^{-1}.$$

## B.2  Algorithm

Here, we present the algorithm used to compute the inverse of a transformation. For the construction of the set $C$, we iterate only over the index set $P$. The set $P$ is constructed do include all points in $G$ that could yield non empty intersections $c_{i',j'}$, thus this is just to speed up the evaluation and equivalent otherwise to the algorithm described in the main part.

**Data:** Image $x' \in \mathbb{R}^{m \times m}$, transform $T$, parameter range $B$, coordinates $i, j$

**Result:** Range for the pixel value $p_{i,j}$.

1 $N \leftarrow \begin{pmatrix} [i-2, i+2] \\ [j-2, j+2] \end{pmatrix}$

2 $\begin{pmatrix} [i'_l, i'_u] \\ [j'_l, j'_u] \end{pmatrix} \leftarrow T_B(N)$

3 $P \leftarrow \left\{ \begin{pmatrix} i' \\ j' \end{pmatrix} \middle| \begin{array}{l} i' \in \text{range}(\lfloor i'_l \rfloor, ..., \lceil i'_u \rceil, 2) \\ j' \in \text{range}(\lfloor j'_l \rfloor, ..., \lceil j'_u \rceil, 2) \end{array} \right\}$

4 $C \leftarrow \left\{ c_{i',j'} := T_B^{-1} \begin{pmatrix} i' \\ j' \end{pmatrix} \cap N \middle| c_{i',j'} \neq \emptyset, (i', j') \in P \right\}$

5 $p_{i,j} \leftarrow [0,1] \cap \left( \bigcap_{c_{i',j'} \in C} q_{i-2,j-2}(c_{i',j'}) \cup q_{i,j-2}(c_{i',j'}) \cup q_{i-2,j}(c_{i',j'}) \cup q_{i,j}(c_{i',j'}) \right)$

**Algorithm 1:** Procedure to calculate the range for the pixel values of the inverse image.

## B.3  Experimental Evaluation

To investigate the impact of refinement on the downstream error estimate we used 20 MNIST images, rotated each with 3 random angles and then proceeded to calculate the inverse. In the calculation, we considered the range $\Gamma_\pm = 10$. We see that a low number of refinements have a large impact on the error but the returns become quickly diminishing. The impact on the run time of a single additional refinement step is negligible.

Figure 4: Interpolation and rounding error $E$ as well as run time for different numbers of refinement steps.

(a) Rotated       (b) Inverse       (c) $10\times$ refined inverse       (d) Original

Figure 5: Computation of the inverse, analogous to Fig. 2, for images from ImageNet [30].

## C  Inverse for Rich Images

INDIVSPT performs poorly on large images, such as those from ImageNet as the inverse computation outlined in Section 6 produces a too large over-approximation of $\boldsymbol{x}$ leading to $E$ estimates of around 40, while manageable value would be $\leq 2$.

Fig. 5 shows the computed inverse for such images. We observe a pattern of artifacts in the inverse, where the pixel value can not be narrowed down sufficiently resulting in the large estimate of $E$. The result of the refined inverse is perfectly recognizable to a human observer (or a neural network), highlighting the promise of the algorithm for future applications.

## D  Experiment Details

### D.1  Details for Section 7.3

To evaluate BASESPT we use the following classifiers. Note that Table 3 in App. E.1 contains results for further datasets:

**MNIST [33]**  We trained a convolutional network consisting of CONV2D$(k, n)$, with $k \times k$ filter size, $n$ filter channels and stride 1, batch norm BN [34], maximum pooling MAXPOOL(k) on $k \times k$ grid, DROPOUT$(p)$ [35] with probability $p$ and linear layers LIN$(a, b)$ from $\mathbb{R}^a$ to $\mathbb{R}^b$.

$$\text{CONV2D}(5, 32), \text{RELU}, \text{BN}$$
$$\text{CONV2D}(5, 32), \text{RELU}, \text{MAXPOOL}(2), \text{DROPOUT}(0.2)$$
$$\text{CONV2D}(3, 64), \text{RELU}, \text{BN}$$
$$\text{CONV2D}(3, 64), \text{RELU}, \text{BN}, \text{MAXPOOL}(2), \text{DROPOUT}(0.2)$$
$$\text{CONV2D}(3, 128), \text{RELU}, \text{BN}$$
$$\text{CONV2D}(1, 128), \text{RELU}, \text{BN}, \text{FLATTEN}$$
$$\text{LIN}(128, 100), \text{RELU}$$
$$\text{LIN}(100, 10)$$

We used data normalization for MNIST and trained for 180 epochs with SGD, starting from learning rate 0.01, decreasing it by a factor of 10 every 60 epochs. No other pre-processing was used.

**FashionMNIST [36]** We trained a ResNet-18 with data normalization. We trained for 180 epochs with SGD with an initial learning rate of 0.01, lowering it by a factor of 10 every 60 epochs.

**CIFAR [32] & GTSRB [37]** We trained a ResNet-18 with data normalization. We trained for 90 epochs with SGD with an initial learning rate of 0.1, lowering it by a factor of 10 every 30 epochs. We resized GTSRB images to $32 \times 32 \times 3$.

**ImageNet [30]** We used the pre-trained ResNet50 from `torchvision`: `https://pytorch.org/docs/stable/torchvision/models.html`.

## D.2  Details for Section 7.4

In Section 7.4 we use the same architectures (with the exception of MNIST, for which we use a ResNet-18) as discussed in App. D.1, however, we trained them to be robust to image transformations (rotation, translation) as well as $\ell^2$ noise.

To train networks that perform well when randomized smoothing is applied, we utilize the training procedure $\text{SMOOTHADV}_{\text{PGD}}$ as outlined in Salman et al. [8]. For each batch of samples we apply a randomized data augmentation, vignetting for rotation, and Gaussian blur. After this prepossessing we then apply $\text{SMOOTHADV}_{\text{PGD}}$ with one noise sample using $\sigma$ and 1 PGD pass (with step size 1; as in Salman et al. [8]) and then evaluate or train on the batch.

The intuition behind the Gaussian blur is that many artifacts, such as the interpolation error are have high frequencies. The blur acts as a low-pass filter and discards high frequency noise. This does not strongly impact the classification accuracy, but drastically reduces the error estimate and therefore the amount of noise that needs to be added for robust classification. The filter is parameterized by $\sigma_b$ and the filter size $s_b$. Formally the filter is a convolution with a filter matrix $F \in \mathbb{R}^{s_b \times s_b}$. Each entry in $F$ is filled with values of a two dimensional Gaussian distribution centered at the center of the matrix and evaluated at the center of the entry. Afterwards the matrix is normalized such that $\sum_{i,j} F_{i,j} = 1$.

In the error estimation and inference we use the same prepossessing as during training.

**MNIST**  For MNIST we use a ResNet-18 (that takes a single color channel in the input layer), which we trained with $\sigma = 0.3$, PGD step size 0.2, batch size 1024, and initial learning rate 0.01 over 180 epochs, lowering the learning rate every 60 epochs. For data augmentation we used rotations in $[-30, 30]$, $[-180, 180]$ degrees and translations of $\pm 50\%$ for each model respectively. For the Gaussian blur we use $\sigma_b = 2.0$ with filter size $s_b = 5$.

**CIFAR-10 & German Traffic Sign Recognition Benchmark (GTSRB)**  For both datasets we use a ResNet-18 trained on $32 \times 32$ images. We use the same pipeline as for ImageNet, but with $\sigma = 0.12$, PGD step size 0.25, batch size 128, and lowered the learning rate every 70 epochs over 500 total epochs. For both datasets we used data augmentation with $\pm 30$ degree rotations. For the Gaussian blur we use $\sigma_b = 1.0$ and $s_b = 5$.

**(Restricted) ImageNet** We trained with a batch size of 400 for 90 epochs using stochastic gradient decent with a learning rate starting at 0.1, which is decreased by a factor 10 every 30 epochs.

We trained either with random rotation (uniformly in $[-30, 30]$ degrees; with bilinear interpolation). Further, we used $\sigma = 0.5$ and PGD step size 1.0, as well as $\sigma_b = 2.0$ and $s_b = 5$.

All training on ImageNet with 4 GeForce RTX 2080 Tis and a 16-core node of aw Intel(R) Xeon(R) Gold 6242 CPU @ 2.80GHz takes roughly 1 hour per epoch and Restricted ImageNet 10 minutes per epoch.

### D.3 Details for Section 7.5

We used the same networks as outlined in App. D.2.

## E Additional Experiments

### E.1 Additional Results for Section 7.3

Table 3: Extended version of Table 1. Evaluation of BASESPT on 1000 images. The attacker used `worst-of-100`. We use $n_\gamma = 1000, \sigma_\gamma = \Gamma_\pm$.

| Dataset | $T^I$ | $\Gamma_\pm$ | Acc. | adv. Acc. | | t [s] |
|---|---|---|---|---|---|---|
| | | | $b$ | $b$ | $g$ | |
| MNIST | $R^I$ | 30° | 0.99 | 0.73 | 0.99 | 0.97 |
| FMNIST | $R^I$ | 30° | 0.91 | 0.13 | 0.87 | 7.98 |
| CIFAR-10 | $R^I$ | 30° | 0.91 | 0.26 | 0.85 | 0.95 |
| GTSRB | $R^I$ | 30° | 0.91 | 0.30 | 0.88 | 8.00 |
| ImageNet | $R^I$ | 30° | 0.76 | 0.56 | 0.76 | 5.43 |
| MNIST | $\Delta^I$ | 4 | 0.99 | 0.03 | 0.53 | 0.86 |
| FMNIST | $\Delta^I$ | 4 | 0.91 | 0.10 | 0.50 | 6.12 |
| CIFAR-10 | $\Delta^I$ | 4 | 0.91 | 0.44 | 0.79 | 0.95 |
| GTSRB | $\Delta^I$ | 4 | 0.91 | 0.30 | 0.63 | 5.17 |
| ImageNet | $\Delta^I$ | 20 | 0.76 | 0.65 | 0.75 | 6.70 |

Table 3 is an extended version of Table 1 and provides results for additional datasets.

Table 4: We first use BASESPT to obtain the certification radius $r_\gamma$ on 30 images and subsequently sample from the parameter space indicated by $\Gamma_\pm = r_\gamma$ and checked whether the certificate holds for them. We use 30 samples and $n_\gamma = 2000$ samples for the smoothed classifier. The last column shows the number of images for which we found violations.

| Dataset | $T^I$ | $\Gamma_\pm$ | median $r_\gamma$ | $r_\gamma$ violated |
|---|---|---|---|---|
| MNIST | $R^I$ | 30° | 28.34 | 0 |
| FMNIST | $R^I$ | 30° | 13.45 | 1 |
| CIFAR-10 | $R^I$ | 30° | 19.16 | 14 |
| GTSRB | $R^I$ | 30° | 20.93 | 0 |
| ImageNet | $R^I$ | 10° | 27.13 | 1 |
| MNIST | $\Delta^I$ | 4 | 1.12 | 0 |
| FMNIST | $\Delta^I$ | 4 | 1.78 | 1 |
| CIFAR-10 | $\Delta^I$ | 4 | 4.76 | 14 |
| GTSRB | $\Delta^I$ | 4 | 2.58 | 0 |
| ImageNet | $\Delta^I$ | 20 | 16.43 | 0 |

Table 5: Same setup as in Table 4, but with circular vignetting.

| Dataset | $T^I$ | $\Gamma_\pm$ | median $r_\gamma$ | $r_\gamma$ violated | $r_\gamma$ violated, no interpolation |
|---------|-------|--------------|-------------------|---------------------|----------------------------------------|
| MNIST    | $R^I$ | $30°$ | 28.34 | 0  | 0 |
| FMNIST   | $R^I$ | $30°$ | 17.07 | 0  | 0 |
| CIFAR-10 | $R^I$ | $30°$ | 11.49 | 10 | 0 |
| GTSRB    | $R^I$ | $30°$ | 25.28 | 0  | 0 |

## E.2 "Certification Radius" of BASESPT

As BASESPT uses Theorem 3.2 to justify the heuristic, this also makes it tempting to use the bound $r_\gamma$ provided by it. However, as the assumptions of Theorem 3.2 are violated it does not formally present a certification radius. Here we investigate if and how much it holds nevertheless. To do this we construct a smoothed classifier $g$ from an undefended base classifier $b$ and calculated the certification radius $r_\gamma$. Subsequently, we sampled 100 new rotated images in the parameter space induced by $\Gamma_\pm = r_\gamma$ and evaluated on them. The results are shown in Table 4. While generally robust, the radius does not constitute a certificate, as we can clearly find violations.

In the context of rotation $R^I$ we add circular vignetting (as we do for DISTSPT and INDIVSPT) to make the behavior closer to a composing transformation. For this experiment, we retrained the same networks, but applied the vignette during training. Results are shown in Table 5 where we can see that this already decreases the number of violations for CIFAR-10 and FMNIST. In a final step we assume knowledge of the attacker parameter $\gamma$ and replace $R^I_\beta \circ R^I_\gamma$ (for the same images) with $R^I_{\beta+\gamma}$ in the evaluation of the classifier, in which case Theorem 3.2 should hold and indeed we don't observe any more violations.

## E.3 Additional Results for Section 7.4

**Beyond Bilinear Interpolation** BASESPT and DISTSPT can directly be applied to image transformations using other interpolation schemes without any adaption. INDIVSPT, however, requires the adaption of the inverse algorithm. While this is generally possible, we consider it beyond the scope of this work.

Table 6 shows the estimated $\epsilon$ (via sampling for $N = 1000$). These values are generally higher than the ones observed for bilinear interpolation, making classification more challenging. For $E$ 1.5 times larger than $\epsilon_{\max}$ we initiated classifier (with 100 images each) and evaluate them in App. E.3. These results indicate that we can obtain similar guarantees as for bilinear although might need to apply higher computation resources ($n_\delta$). For (R)ImageNet taking $E$ to 1.5 the maximal error this error becomes too large to handle. Realistically one could use an $E$ between 1.84 and 2.76 depending on the desired $q_E$.

Table 6: The observed $\epsilon$ values. Showing the maximum $\epsilon_{\max}$ and the 99-th percentile $\epsilon_{99}$.

| Dataset | $\epsilon_{\max}$ | $\epsilon_{99}$ |
|---------|-------------------|-----------------|
| MNIST    | 0.44 | 0.29 |
| CIFAR-10 | 0.88 | 0.74 |
| ImageNet | 1.84 | 1.52 |

Table 7: Results for DISTSPT on bicubic interpolation. $\sigma_\gamma = \Gamma$.

| Dataset | $T^I$ | $E$ | $\Gamma_\pm$ | Acc. | | $r_\gamma$ percentile | | | T [s] | $n_\gamma$ | $n_\delta$ |
|---------|-------|-----|--------------|------|------|---------------------|---------------------|---------------------|--------|-----------|-----------|
| | | | | $b$ | $g$ | $25^{\text{th}}$ | $50^{\text{th}}$ | $75^{\text{th}}$ | | | |
| MNIST     | $R^I$ | 0.66 | $30°$ | 0.99 | 0.99 | $30.00^\dagger$ | $30.00^\dagger$ | $30.00^\dagger$ | 8.30   | 200 | 2000  |
| CIFAR-10  | $R^I$ | 1.11 | $30°$ | 0.72 | 0.54 | 1.87            | 18.37           | $30.00^\dagger$ | 24.67  | 50  | 10000 |
| RImageNet | $R^I$ | 1.84 | $30°$ | 0.72 | 0.68 | 14.34           | $30.00^\dagger$ | $30.00^\dagger$ | 169.25 | 50  | 4000  |
| RImageNet | $R^I$ | 2.76 | $30°$ | 0.72 | 0.00 | -               | -               | -               | 151.88 | 50  | 4000  |

### E.4 Audio Volume Change

To show that our method can be used beyond image transformation we showcase an adaption to audio volume changes. The volume of an audio signal can be changed by multiplying the signal with a constant. In order to change the signal $\boldsymbol{x}$ by $\beta$ (measured in decibel $[\beta] = $ dB) we multiply $\boldsymbol{x}$ by $10^{\beta/20}$. Thus the transformation is $\psi_\beta(\boldsymbol{x}) := 10^{\beta/20} \cdot \boldsymbol{x}$, which composes:

$$\psi_\beta \circ \psi_\gamma(\boldsymbol{x}) = 10^{(\beta+\gamma)/20} \cdot \boldsymbol{x} = \psi_{\beta+\gamma}(\boldsymbol{x}).$$

In practice such signals are stored in final precision, e.g. 16-bit, thus potentially introducing rounding errors, with an $\ell^2$-norm bound by $E$. If this is ignored BASESPT can be applied to obtain guarantees. Otherwise, DISTSPT and INDIVSPT can be used to obtain sound bounds.

To evaluate this we use the speech commands dataset [38], consisting of 30 different commands, spoken by people, which are to be classified. The length of the recordings are one second each. We use a classification pipeline that converts audio wave forms into MFCC spectra [39] and then treats these as images and applies normal image classification. We use a ResNet-50, that was trained with Gaussian noise, but not SMOOTHADV$_{\mathrm{PGD}}$. We apply the noise before the waveform is converted to the MFCC spectrum.

For DISTSPT we estimate $E$ to be 0.005 with the parameters $\alpha_E = 0.05, \sigma_\gamma = 3$ and $\Gamma = 3$. On 100 samples, the base classifier $f$ was correct 94 times, and the smoothed classifier $g$ 51 times for $r_\gamma$ of 0.75, 1.96 and 3.12 for the $25^{\mathrm{th}}$, $50^{\mathrm{th}}$, $75^{\mathrm{th}}$ percentile respectively, corresponding to $\pm 1.09, \pm 1.25$ and $\pm 1.43$ dB. At $n_\gamma = 100$ and $n_\epsilon = 400$ the average certification time was $138.06s$.

To investigate INDIVSPT we use $\sigma_\gamma = 0.85, \Gamma = 1.05$. For 92 out of 100 perturbed audio signals to compute $\epsilon$. We obtained $\epsilon_{\max} \leq 0.0055$ and for 68 an $\epsilon \leq 0.005$, which together with our results for DISTSPT suggests the applicability of the method. For each signal we used 100 samples for $\beta$. For cases with $\epsilon_{\max} > 0.0055$ we in fact observed $\epsilon_{\max} \gg 0.0055$, as here many parts of the signal were amplified beyond the precision of the 16-bit representation and clipped to $\pm 1$. This makes the information unrecoverable and sound error bound estimates large.

Table 8: Maximum observed errors and without gaussian blur (G) and without vignetting (V).

| Dataset | Both | -V | -G | -V-G |
|---|---|---|---|---|
| MNIST | **0.36** | **0.36** | 2.47 | 2.51 |
| CIFAR-10 | **0.51** | 6.08 | 2.66 | 18.17 |
| ImageNet | **0.91** | 70.66 | 9.25 | 75.69 |

Table 9: Correct classifications and by the model and verifications by DeepG [11], with and without vignetting (V), out of 100 images.

| Model | Correct | [11] | [11]+V |
|---|---|---|---|
| MNIST | 98 | 86 | 87 |
| CIFAR-10 | 74 | 65 | 32 |
| CIFAR-10+V | 78 | 63 | 23 |

## F   Further Comparison and Ablation

To show that the vignette and Gaussian blur are essential to our algorithm we perform a small ablation study. Table 8 shows the maximal error observed when sampling as in DISTSPT. We use the same setup as in Section 7.4, but with 10000 samples for ImageNet.

Both, vignetting and Gaussian blur reduce the error bound significantly for DISTSPT and INDIVSPT. On CIFAR-10 and ImageNet vignetting is very impactful because the corners of images are rarely black in contrast to MNIST. Li et al. [13] uses vignetting for the same reason. Without either of the methods bounding the error would not be feasible.

For INDIVSPT vignetting is crucial, even for MNIST, as we can make no assumptions for parts that are rotated into the image. Thus we need to set these pixels to the full $[0, 1]$ interval (see Fig. 2). Without Gaussian blur the certification rate drops to 0.11.

Further, we extend this comparison to related work: We extended Balunovic et al. [11] (Table 1 in their paper) to include vignetting. The results are shown in Table 9. We also retrained their CIFAR-10 model with vignetting (CIFAR-10+V) for completeness. While vignetting on MNIST slightly helps

(+1 image verified) on CIFAR-10 it leads to a significant drop. Including Gaussian blur into [11] would require non-trivial adaption of the method. However, we implemented this for interval analysis (on which their method is built) and found no impact on results.