[Reviews · NeurIPS 2020]

Review 1

Summary and Contributions: The authors present a generalization of randomized smoothing to arbitrary parameterized transformations: rotations or translations in the image domain, and scaling in the audio domain. They provide three flavors of this: 1) a heuristic defense that yields no certificate, 2) a distributional guarantee where certification is applied based on parameters inferred from the training set, and 3) an online defense which can provide a heuristic defense and certification for individual test examples. Much care is taken to account for the interpolation error induced by image representations. Experiments are performed on MNIST/CIFAR/ImageNet demonstrate that this technique is scalable and provides nontrivial bounds.

Strengths: Robustness certification to adversarial examples typically considers the case where an adversary is only allowed to add L_p bounded noise, and typically is not scalable, except for the randomized smoothing approaches. This work extends those results by considering adversaries that can perform any parameterized transformation and is, as expected, scalable to large images/networks. The consideration of interpolation error and care taken to apply standard interval-analysis techniques to inversion of rotations is novel and interesting. The experimental section is very thorough and accurately demonstrates the scalability/flexibility of the presented technique. The authors provide a fair and honest discussion of the drawbacks of their approach, including a broad characterization for datasets for which their method will be weak.

Weaknesses: The primary theoretical contribution, while novel, is a fairly incremental improvement from the original analysis presented by Cohen et al. While the authors claim that this method is easily generalizable to arbitrary parameterized transformations, they restrict their attention to transformations that are already well-studied, such as rotations and translations, omitting transformations such as spatial transformations (stAdv/Wasserstein AdvEx) that have high parameter dimension. Indeed, on such highly parameterized transformations, it is unclear how the inverse may be calculated efficiently (i.e., could the interval splitting approach on line 145 work?). Further, while the authors offer discussions on competing methods, they do not explicitly compare against these methods on CIFAR. Finally, it seems like many tricks were applied to make their method work well (e.g. vignette, gaussian blur); it would be interesting to see results without the highly optimized flavor of the proposed technique.

Correctness: The theoretical results seem correct and fairly standard. The experimental section is quite thorough and appears standard and well documented. Code was provided along with this submission.

Clarity: I found the paper fairly hard to follow in general. Better signposting throughout would help more clearly organize the paper. For example, more explicitly discussing the need for a pre-smoothed classifier in 4.1 would have been helpful. My personal preference is for exposition using symbols/formulas rather than examples, as in section 5.3. There are also several small typos (e.g.) line 117: E>=eps(...) should be ||E|>=eps(...) line 156: RHS should have a max over i line 285: bellow -> below

Relation to Prior Work: The authors clearly describe previous techniques in robustifying/certifying neural networks to rotations/translations. As mentioned in the weaknesses section, it would be helpful to have direct comparison of their experimental results to previous techniques using all the tricks (vignette/gaussian blur) they applied as well.

Reproducibility: Yes

Additional Feedback:


Review 2

Summary and Contributions: This paper introduces a generalization of randomized smoothing to derive a provable defense against parametrized image transformations. It introduces robustness certification guarantees both at the distributional level (whole dataset) and individual level (per image). They achieve provable distributional robustness to rotation based adversarial attacks.

Strengths: The paper introduces a generalization of randomized smoothing to derive individual and distributional robustness certificates that can scale to the size of Imagenet. In addition, they derive a novel mechanism to handle the interpolation errors resulting from image transformations.

Weaknesses: The paper suffers from the same problems as randomized smoothing. For inference, the number of samples that need to be computed can be very large. For CIFAR, errors due to interpolation tend to be high.

Correctness: The theoretical claims and empirical methodology seem to be correct.

Clarity: The paper is very well written and easy to follow.

Relation to Prior Work: The work clearly discusses the prior work and the pros and cons of this work compared to previous works.

Reproducibility: Yes

Additional Feedback:


Review 3

Summary and Contributions: In this paper, the author presents a certified defense method against adversarial image transformations. The presented method is based on random smoothing thus can scale to large DNN models and datasets.

Strengths: 1. This method can be applied to a wide range of application domains and image transformations. 2. This method can scale to large datasets and complex DNN models. 3. The author presents defense solutions for both distributional and individual settings.

Weaknesses: 1. Most of the results lack a comprehensive comparison with previous methods (only some brief descriptions in comparison to other work). The author should provide more comparisons between their method and previous works on certified defense methods against adversarial transformation. 2. Section 5.3 should be simplified.

Correctness: The claims are correct.

Clarity: The author should improve writing.

Relation to Prior Work: It is clearly discussed

Reproducibility: Yes

Additional Feedback: 1. One merit of this method is the scalability on large-scale datasets. Does this come from the use of random smoothing? 2. From the paper, it seems that the current method can be only applied to the case with one specific adversarial transformation? Can this method be extended to the context where multiple transformations exist?


Review 4

Summary and Contributions: In this work, the authors extend the probabilistic robustness certification argument of randomized smoothing (RS) to a few different domains. The most interesting, in my opinion, being that of parameterized transformations. Doing so comes with some technicalities in terms of rounding that the authors explain clearly and deal with. They also perform a similar argument to this in order to get a bound on the error arising from any point in the distribution.

Strengths: The paper extends the randomized smoothing argument to a parameterized function that can handle such transformations as rotation and translation. Though other methods give guarantees for these transformations (in the form of linear propagation or other over approximations), this paper is, to my knowledge, the first paper to extend the randomized smoothing argument in this way. Given that it is always valuable to have an arsenal of models to properly capture the invariances of deep neural networks I think this work has potential to be impactful.

Weaknesses: I am quite keen to see how figure 2 looks when applied to context-rich RGB images such as CIFAR and ImageNet. I took a quick glance at the appendix and did not find any examples there. The authors provide a similar argument to SPT but for the data distribution, however, this guarantee must hold over the entire, unknown data distribution and so it seems quite laborious to compute. However, to their credit, I still think that such a measure is informative even if the authors are unable to compute it with rigorous and decent statistical guarantees in practice.

Correctness: The empirical methodology is sound; however, I was unable to check the proofs of this paper entirely which I will do before the final decision.

Clarity: The paper is very well written and I appreciate the way the authors break down the way that they deal with the inverse computation and rounding issues.

Relation to Prior Work: The authors clearly survey the literature, and to the best of my knowledge cite the most prominent and recent papers in certifying geometric transformations.

Reproducibility: Yes

Additional Feedback: I would like to thank the authors for responding to my query about visualizing some of their perturbations. I think the paper makes an interesting and novel contribution and have given it another couple reads in the mean time. I am convinced at the novelty and position of the work, yet I was still unable to find enough time to check the full details of the mathematical derivation for correctness; however, I am increasing my confidence score on the basis of fully understanding this paper's position in the literature.

[Author Response · NeurIPS 2020]

We thank the reviewers for their positive and helpful feedback. We are encouraged that they find our method scalable
(R1, R3), widely applicable (R3) and our handling of interpolation errors (R1, R2) and evaluation extensive (R1). In a
thorough revision we will include the reviewers' suggestions. Specifically we will (i) improve the outline of the paper
(R1), (ii) clarify in Sec. 4.1 the need of a certifiably robust classifier $h_E$ (R1) and (iii) add to Sec. 5.3 the explanation to
calculate the inverse of the method in general (R1, R3). Below we answer the individual questions.

**R1; Can you handle high dimensional transformations (stAdv/Wasserstein AdvEx)?** BASESPT and DISTSPT can
be applied directly to highly parametrized transformations like $\ell^2$ bound vector field transformations ($\sum_p \|v_p\|_2^2 \leq \tau$,
$v_p$ denotes the displacement of pixel $p$, similarly to stAdv). However, INDIVSPT can only be applied for small $\tau$ as
calculating a tight inverse is challenging, because interval splitting (L145) is not feasible. A not-tight inverse can be
obtained by the relaxation that $\|v_p\|_2^2 \leq \tau$ for all pixels $p$ individually. The Wasserstein transformation does not use
interpolations, thus BASESPT is sound (cf. [arXiv:1910.10783]) and DISTSPT/INDIVSPT are not needed.

**R1; Can you show results without vignetting and Gaussian blur?**
Yes. The results for BASESPT (Sec. 6.1) were obtained without these
techniques (we will clarify this).

| Dataset | Paper | -V | -G | -V-G |
|---|---|---|---|---|
| MNIST | **0.39** | **0.39** | 2.38 | 2.49 |
| CIFAR10 | **0.77** | 4.64 | 2.48 | 21.04 |
| ImageNet | **0.95** | 70.66 | 9.25 | 75.69 |

Table 1: Maximum observed errors and without gaussian blur (G) and without vignetting (V).

For DISTSPT (Sec. 6.2) the error estimates without vignetting or
Gaussian blur are shown in Table 1. The setup was the same as in
Sec. 6.2, but for ImageNet we used 10000 instead of 700000 sam-
ples. Both, vignetting and Gaussian blur improve the error bound
significantly. On CIFAR10 and ImageNet vignetting is very impactful
because the corners of images are rarely black. [13] uses vignetting for
the same reason. Gaussian blur helps to shrink the errors for images
particularly sensitive to interpolation, i.e. a chess board.

For INDIVSPT vignetting is crucial, even for MNIST, as we can make no assumptions for parts that are rotated into
the image. Thus we need to set these pixels to the full $[0, 1]$ interval (see Fig. 2 in the paper and Fig. 1 here). Without
Gaussian blur, the verification rate drops from 99.6% to 11.6% on MNIST. We will include details in an appendix.

**R1, R3; Can you compare to prior work more extensively?**
**R1; How would vignetting and Gaussian blur benefit them?**
Yes, we extended [11] (Table 1 in their paper) to include vignetting.
The results are shown in Table 2. We also trained a CIFAR10 model
with vignetting (CIFAR10+V) for completeness. While vignetting
on MNIST slightly helps (+1 image verified) on CIFAR10 it leads to
a significant drop. Including Gaussian blur into [11] would require

| Model | Correct | [11] | [11]+V |
|---|---|---|---|
| MNIST | 98 | 86 | 87 |
| CIFAR10 | 74 | 65 | 32 |
| CIFAR10+V | 78 | 63 | 23 |

Table 2: Correct classifications and by the model and verifications by DeepG [11], with and without vignetting (V), out of 100 images.

non-trivial adaption of the method. However, we implemented this for
interval analysis (on which their method is build) and found no impact
on results. We will extend the our discussion (L274ff., L312ff.) similar
to this discussion and more directly compare with our CIFAR10 results
(App. E). Other related work is either in a fundamentally different setting or subsumed by the discussed works.

**R3; Does your scalability originate from randomized smoothing?** Yes. Meth-
ods relying on convex relaxation (e.g., [11]) for neural network verification suffer
from accumulation of overapproximation and the slow runtime. While we still use
interval analysis for DISTSPT and INDIVSPT to bound the interpolation error,
we circumvent both problems by verifying with randomized smoothing.

(a) Rotated  (b) Original

**R3; Can your method be applied to combinations of transformations?** Yes,
Theorem 3.2 can be applied to *composable transformations*, that is $\psi_\beta \circ \psi_\gamma = \psi_{\beta+\gamma}$
$\psi_{\beta+\gamma}$ (L92-93). In Sec. 4.1 we consider the case where this holds approximately.
Unfortunately, as rotations $R$ and translations $T$ do not commute, $\psi_\beta := R_{\beta_1} \circ$
$T_{\beta_2,\beta_3}$ is not composable, i.e. $\psi_\beta \circ \psi_\gamma \neq \psi_{\beta+\gamma}$ (L309-311).

(c) Inverse

**R4; Can you show Figure 2 for a context rich RGB image?** Yes, see Fig. 1
for an example from ImageNet. As outlined in L284-285 there are images with
very large error ($> 50$). This error stems from the regions where the inverse
algorithm can't determine strong constraints on the pixel value, visible as the
circular pattern. Since the submission we have investigated improvements for
such images and found partial success by replacing the circular vignette with an
adaptive mask based on the local quality of the inverse. We will supplement the
paper with example images and further discussion.

(d) $10\times$ refined inverse

Figure 1: Image with high error. In (c) & (d): Lower and upper bound.

[Meta-Review · NeurIPS 2020]

Reviewers found the idea of the paper and the empirical results interesting and significant. In the final draft, authors should address the comments raised by the reviewers.